# The COVID-19 Pandemic Impact of Hospital Wastewater on Aquatic Systems in Bucharest

Alina Roxana Banciu [1,†] , Luoana Florentina Pascu [1,†] , Dragos Mihai Radulescu [1], Catalina Stoica [1],
Stefania Gheorghe [1] , Irina Lucaciu [1], Florin Valentin Ciobotaru [1,2], Laura Novac [1,3], Catalin Manea [1]
and Mihai Nita-Lazar [1,*]

[1]   National Research and Development Institute for Industrial Ecology-ECOIND, 57-73 Drumul Podu
      Dambovitei, 060652 Bucharest, Romania; alina.banciu@incdecoind.ro (A.R.B.); ecoind@incdecoind.ro (L.F.P.);
      dragos.radulescu@ecoind.ro (D.M.R.); catalina.stoica@incdecoind.ro (C.S.);
      stefania.gheorghe@incdecoind.ro (S.G.); irina.lucaciu@incdecoind.ro (I.L.); florin.ciobotaru@ecoind.ro (F.V.C.);
      laura.novac@ecoind.ro (L.N.); catalin.manea@incdecoind.ro (C.M.)
[2]   Faculty of Geography, University of Bucharest, Nicolae Balcescu Boulevard, no. 1, 010041 Bucharest, Romania
[3]   Faculty of Biotechnology, University of Agronomic Sciences and Veterinary Medicine of Bucharest, 59 Marasti
      Boulevard, 011464 Bucharest, Romania
*     Correspondence: mihai.nita@incdecoind.ro
†     These authors contributed equally to this work.

**Abstract:** The COVID-19 pandemic reshaped the global response to a pandemic, including the
way of using chemical compounds such as disinfectants and antibiotics. The large-scale use of
antibiotics and disinfectants during the COVID-19 pandemic caused environmental pressure not
only due to the chemicals themselves but also due to their effect on bacterial communities, inducing
resistance to chemicals and changing the population structure of bacterial communities, especially
in aquatic environments. The dissemination of fecal bacteria, including antibiotic-resistant bacteria,
and pathogens from hospital wastewater into the environment, via wastewater treatment plants
(WWTPs), triggered the premises of a major public health issue. Rivers flowing through cities are
natural streams for WWTP discharges, and they directly bear the impact of anthropic activities,
disseminating domestic and industrial pollution over large areas. The aim of the present study was to
assess the microbiological bacterial structure of municipal and hospital wastewaters as well as their
impact on natural streams, covering the pre-to post-COVID-19 pandemic period of time. The results
indicated that the COVID-19 pandemic had a direct impact on hospital wastewater microbiological
quality and the environment due to an excessive use of antibiotics and disinfectants. In addition, the
constant presence of antibacterial compounds increased the rate of bacterial selection and induced
population structural changes in the bacterial communities from aquatic systems.

**Keywords:** bacteria; environment; aquatic ecosystem; COVID-19 pandemic

## 1. Introduction

Rivers flowing through cities are often used as receiving bodies for treated and un-
treated urban wastewaters. Some of these streams are amongst the most persuasive exam-
ples of ecosystems disturbed by anthropic activities related to uncontrolled discharges [1].
The impact of released wastewater on a receiving water body has been directly linked
to the population size of the city, the input concentrations of pollutants such as personal
care products and household chemicals, and the technological level and efficiency of the
wastewater treatment plant (WWTP) as well as the capacity of the stream to dilute the
WWTP discharge.

Municipal wastewater has a very complex pollutant composition, including household
chemicals, pharmaceutical compounds from the hospital care units, and pollutants from
industrial workshops and commercial and service activities [2]. The pollutant complexity

affects not only microbial communities in the biological treatment step associated with WWTPs but also microbial communities from aquatic systems, which are one of the key players in the biogeochemical cycling of organic matter and nutrients in nature. The impact on the microbial population has been directly linked to the resilience, recovery, and sustainability of ecosystem health as well as public health. Investigating the fecal population dynamic in a WWTP system and evaluating the resistance and resilience of microbial communities are essential for understanding how the aquatic environment could be affected by controlled or uncontrolled wastewater discharges [1]. The presence of fecal bacteria, including pathogens, in various ecosystems could be a sign of major pollution and a warning signal for public health, especially when bacteria develop adaptation and resistance mechanisms to pharmaceutical compounds, which are massively used in hospitals [3]. Recently, Ramirez-Coronel and collaborators showed that wastewaters from hospitals and medical centers could play a significant role in spreading pathogenic bacteria and subsequently infectious diseases over large areas of the aquatic environment [4]. In addition, hospital wastewaters contain larger loads of infectious microorganisms compared to urban wastewaters, and unfortunately, they can pass and dissipate the bacterial resistance to some pharmaceutical compounds, by lateral gene transfer, to other microorganisms in urban wastewaters and environments [5]. Bacterial resistance to pharmaceuticals is directly linked to the degree of their harmful potential effect on the environment and public health. Hospital wastewaters contain pharmaceutical compounds such as antibiotics and biocides that are not completely removed by a WWPT's biological and physico-chemical treatment processes [6]. Unfortunately, most of the pharmaceutical compounds end up in natural streams, polluting the environment and triggering antibiotic and/or biocide resistance in microorganisms from new natural areas.

The massive increase in antibiotic and biocide production generated a rise in the bacterial antibiotic resistance phenomenon, creating a Red Queen effect. This effect translates into controlling bacterial populations by a larger production and uptake of pharmaceuticals or by new antibiotic compound designs, with more and more complex chemical structures, which in fact have enhanced the antibiotic resistance mechanism in bacteria. It was predicted that there will be an antibiotic production increase by 67% from 2020 to 2030 [7], which will amplify the antibiotic resistance phenomenon. The World Health Organization (WHO) predicted that antibiotic resistance will the one of the top three major public health threats [8].

It is particularly concerning when pathogenic bacteria from the ESKAPE group (*Enterococcus faecium*, *Staphylococcus aureus*, *Klebsiella pneumonia*, *Acinetobacter baumanii*, *Pseudomonas aeruginosa*, and *Enterobacteriaceae*) acquire antibiotic resistance genes by horizontal gene transfer or spontaneously when constantly exposed to antibiotics [9].

Hospital wastewater can be 5- to 15-fold more toxic than domestic wastewater because the majority of emerging contaminants are capable of inhibiting biological activities essential for an efficient WWTP. They escape from WWTPs, and their widespread occurrence in natural streams can be detrimental to the health of humans and wildlife [6,10]. Hospitals are playing a significant role in emerging and spreading pathogenicity characteristics, along with domestic households, because most antibiotics used in clinical settings as well as their antibiotic-resistant bacteria end up in municipal and industrial wastewaters.

Wastewaters from healthcare settings have been reported to be associated with higher antibiotic concentrations and a complex degree of resistance compared with other metropolitan areas [11]. Unfortunately, Romanian legislation has not updated the threshold of microbiological loads allowed to be released into municipal wastewaters. Therefore, antibiotics and subsequently bacterial populations provide selective environmental pressure for developing resistance to chemical compounds. In spite of the fact that wastewaters have been collected and treated in WWTPs, the environmental front-line protection against chemical and pharmaceutical compounds used in hospitals, they have not been highly efficient in removing all emerging pollutants. Therefore, WWPTs become major hubs for microorganisms and pharmaceutical chemical compounds accumulation as well as antibi-

otics resistance genes exchange and dissemination between various bacteria, increasing the number of antibiotic-resistant bacterial strains [11,12].

The antibiotic-resistant bacteria dissemination from WWTP to the environment has attracted the attention of the United Nations, which in 2019 published a report stating "the natural environment becomes a reservoir of antibiotic residues, resistant pathogens and other molecules with antimicrobial properties that increase the spread of antibiotic-resistant genes into the microbial environment" [13].

The COVID-19 pandemic had a direct impact on healthcare systems and environment [14] due to an excessive use of disinfectants and antimicrobial agents in hospital units. A massive increase in antibiotics prescribed during clinical therapies has been linked with high concentrations of antibiotics in hospital wastewaters. This observation was backed up by the human body antibiotic processing rate, when only 15% to 30% of antibiotic amounts, recommended for clinical therapy, could be absorbed and metabolized by humans. Subsequently, a significant percent of antibiotics taken by patients was excreted in wastewater, as still active pharmaceutical compounds, through feces and bodily liquids [15]. The association between antimicrobial resistance and the COVID-19 pandemic is slowly emerging as a major health threat, which is based on an imbalanced approach to antimicrobial consumption. Thus, it is highly probable that municipal wastewater and WWTPs efficiency could be affected by the coronavirus outbreak. The antibiotic resistance genes evaluation in hospital units showed a wide presence and diversity of resistance genes belonging to all resistance classes. The antibiotic resistance genes presence increased up to 40% during the COVID-19 pandemic compared with the pre-COVID-19 pandemic. The change in antimicrobial resistance patterns was generated by widespread antibiotics usage during the COVID-19 pandemic. Pathogenic bacteria gradually develop antibiotic resistance mechanisms that altered the population structure and dynamics of bacterial communities [12].

The COVID-19 pandemic started in China in early December 2019 and quickly spread around the world, triggering harsh social measurements such as lockdowns for large periods of time to maintain adequate social distancing measures in the effort to contain the viral spread. In Romania, the epidemic has gradually spread from April /May 2020 and it ended in the second half of 2022, but its effects are still present at the social, healthcare system and environmental levels.

The aim of this study was to monitor the microbiological wastewater quality and its bacterial populational structure as well as their impact on natural aquatic ecosystems in a period covering the pre- to post-COVID-19 pandemic. The area of interest included wastewaters from hospital units and households located in Bucharest, Romania, as well as their treatment and discharge into the environment. Bucharest city reported the most cases of COVID-19 infections during the pandemic. The processes of urbanization and industrialization have increased the volume of wastewater processed by the municipal WWTP then discharged into Dambovita river—the main source of drinking water for the population of Bucharest. The source of this river is in the Fagaras Mountains, and it flows south. The Dambovita river passes through Bucharest and flows into the Arges River at 258 km from its source. Unfortunately, the river could also transport contaminants escaped from WWTP over large areas, which could have a significant impact on human health and aquatic ecosystem sustainability.

The pre- to post-COVID-19 pandemic assessment of microbiological quality included coliform bacteria and enterococci. Coliform bacteria are a group of aerobic and facultative anaerobic bacteria, and they are Gram-negative microorganisms which can be present in the aquatic environment affected by anthropic activities. This special group of bacteria, the Enterobacteriaceae family, typically belongs to a fecal matter microbiota from human and animals. The presence of total coliform bacteria in water indicates an environmental contamination, while the presence of fecal coliform bacteria indicates a fecal contamination [16]. Moreover, the presence of enterococci indicates an anthropic activity and a discharge of contaminated wastewater.

## 2. Materials and Methods

### 2.1. Sampling

Seven sampling points were selected in Bucharest. Three sampling points were represented by three infectious diseases hospitals from where only effluent wastewater samples were collected. Two sampling points belonged to a municipal WWTP from where influent and effluent wastewater samples were collected. The last two sampling points were on a WWTP natural stream, called Dambovita River; more specifically, they were situated 200 m upstream and downstream from the WWTP discharge site into the natural stream.

The sampling period covered the pre-COVID-19 pandemic (the year of 2019), COVID-19 pandemic (between 2020 and 2021) and post-COVID-19 pandemic (from 2022 to 2023) periods of time. The sampling campaigns had a frequency of at least two sampling collections per season, at there were least 8 sampling campaigns per year between the year of 2019 and 2023.

The water samples were collected in replicate from the seven sampling points in 1 L sterile containers, according to ISO 19458 [17], and then transported and stored at 4–5 °C for a maximum of 24 h before analyses.

### 2.2. Microbiological Analysis

The densities of total coliform and fecal coliform bacteria from wastewater were performed in duplicate by the Most Probable Number (MPN) method (IDEXX) [18]. Briefly, a volume of 100 mL from each sample and its decimal dilution ($10^{-1}$–$10^{-6}$) were homogenized with Colilert-18 medium (Idexx Laboratories, Inc, Westbrook, ME, USA). The mixed Colilert-18 medium was dispensed in special Idexx bags with 49 large wells and 48 small wells, using one bag per each microbiological indicator to be analyzed. These samples were incubated at 36 ± 2 °C for 18–22 h for total coliform bacteria quantification or at 44.5 °C for 18–22 h for fecal coliform bacterial quantification. Bacterial strains used as positive controls were *Escherichia coli* (ATCC 25922) (Sharlab SL, Barcelona, Spain), *Citrobacter freundii* (ATCC 8090) (Thermo Fisher Scientific, Waltham, MA, USA) and *Klebsiella aerogenes* (ATCC 13048) (Sigma-Aldrich, Saint Louis, MI, USA). *Enterococcus faecalis* (ATCC 29212) (Sigma-Aldrich, Saint Louis, MI, USA) was used as the negative control. The results of the Idexx Method were calculated using the Idexx table (x/y axis system), and they were expressed in MPN/100 mL.

Enterococci densities were analyzed by the membrane filtration method [19] with one presumption test and one confirmation test. Briefly, 100 mL of sample and its dilutions ($10^{-1}$–$10^{-6}$) were filtrated through a membrane with a 45 μm pore diameter; then, it was placed on Slanetz and Bartley agar (Oxoid, Basinstoke, England) plates and incubated at 36 ± 2 °C for 48 h. The membranes that showed brown colonies specific to Enterococcus after the first incubation were transferred onto Bile Aesculine Agar (Sharlab SL, Barcelona, Spain) and were incubated at 44 °C for 2 h. All colonies that metabolized esculin from the confirmation medium were counted as Enterococcus, and the result was calculated per 100 mL of sample. *Enterococcus faecalis* (ATCC 29212) (Sigma-Aldrich, Saint Louis, MI, USA) was used as the positive control and *Escherichia coli* (ATCC 25922) (Sharlab SL, Barcelona, Spain) was used as the negative control. At the same time, a blank control (only sterile distilled water) was used for both quantification methods. The results of the membrane filtration method were expressed in colony-forming units (CFU) per 100 mL.

### 2.3. Statistically Analysis

The statistical data were obtained by ANOVA tests.

## 3. Results and Discussion

### 3.1. Quantification of Hospital Wastewater Microbial Load

COVID-19 infections had a worldwide spread, triggering a huge impact on human society by changing the sanitary and social behavior related to unusual and sudden lockdowns as well as new worldwide travel policies [20]. The lockdown sanitary measures

came in parallel with a massive surge in using pharmaceutical compounds, especially biocides and antibiotics. Unfortunately, the rise of pharmaceutical compounds production and their overestimated consumption put extra pressure on the industrial and domestic wastewater treatment system and subsequently on the environment. Pharmaceutical compounds' increased usage had a huge impact on microbial communities from the environment, including those involved in biological wastewater treatment [21]. Changes in the microbial community structure during the COVID-19 pandemic were very noticeable when the total coliform bacterial density from hospital wastewater significantly decreased more than 50% compared to the pre- and post-COVID-19 pandemic (Figure 1a). The same pattern was also observed for enterococcus bacterial populations isolated from hospital wastewater during the COVID-19 pandemic (Figure 1b).

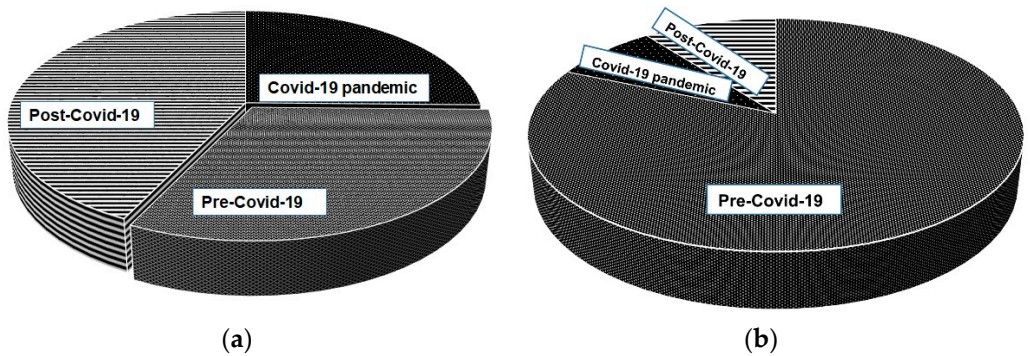

(**a**)　　　　　　　　　　　　　　　　　　　　(**b**)

**Figure 1.** Variation of coliform bacteria density (**a**) and enterococci density (**b**) in hospitals wastewater covering from pre- to post-COVID-19 pandemic period of time—Bucharest, Romania. All results represented average values from at least two independent experiments. Each experiment was repeated twice ($p < 0.05$).

The reduction in total coliform bacteria and enterococcus during the COVID-19 pandemics could be explained by an increase in antibiotics and biocides usage in hospitals compared to pre- and post-COVID-19 pandemic time periods. The usage of large quantities of pharmaceutical, especially biocides and antibiotics, cleared some bacterial strains which could not adapt to the new hospital environmental conditions. Unfortunately, in some situations, the massive antibiotics usage enhanced bacterial antibiotic resistance mechanisms, generating antibiotic-resistant bacteria with serious consequences on health systems and the environment. Research studies during the COVID-19 pandemic periods of time showed an inverse relationship between the massive use of antimicrobial compounds and a low pollution level as a result of a significant decrease in human activities and wastewater discharges [2,12,22].

The collection and treatment of hospital wastewater has been proved to be essential in cutting down the chemical and bacterial loads. If not treated, wastewaters could spread important bacterial loads and pharmaceutical compounds initially to the municipal sewage system and then to the environment. As a consequence, untreated hospitals wastewater could disturb the balance of microbial communities from municipal WWTP by changing their dynamics, spreading antibiotic resistance genes. This could have an impact on the WWTP treatment efficacy with further consequences in polluting the environment via natural stream acceptors of WWTP effluents [23,24]. The total coliform bacteria load values returned during the post-COVID-19 pandemic to a similar value range recorded before the emergence of COVID-19 or the pre-COVID-19 pandemic (Figure 1a). In the case of enterococcus, they still have not matched the pre-COVID-19 pandemic period, remaining at the level of the COVID-19 pandemic period (Figure 1b).

*3.2. Microbial Load from WWTP*

Research studies showed that during the COVID-19 pandemic the bacterial antibiotic resistance also increased, especially for Gram-negative bacteria such as fecal coliforms and

*E. coli*. A slower acquiring antibiotic resistance rate was reported for Gram-positive bacteria such as *Enterococcus* compared to Gram-negative bacteria [25]. A slow acquired antibiotic resistance of Gram-positive enterococcus could explain the low *Enterococcus* density observed during the post-COVID-19 pandemic, which was similar to the COVID-19 pandemic level. In contrast, the acquired resistance of Gram-negative bacteria (total coliforms and *E. coli*) to a large number of pharmaceutical compounds used during the COVID-19 pandemic could explain the massive increase in coliform loads during the post-COVID-19 pandemic, which were even above the pre-COVID-19 pandemic levels. Overall, the difference between the densities of bacterial communities could be explained by a higher acquisition rate of antibiotic resistance genes in total coliform bacteria compared to enterococci.

The total coliform bacteria density detected in sewage wastewater WWTP influent analyzed from the post-COVID-19 pandemics, having approximatively $35 \times 10^6$ MPN/100 mL, reached the pre-COVID-19 pandemic levels. The total coliform bacteria density from WWTP wastewater influent drastically decreased up to 3.5-fold, $10 \times 10^6$ MPN/100 mL during the COVID-19 pandemics (Figure 2) compared to pre- and post-COVID-19 pandemic periods of time. The overall density of total coliforms from WWTP influents decreased during the period of the COVID-19 pandemics, but it remained almost the same during the pre- and post-COVID-19 pandemic period. The total coliform bacteria densities from effluents registered an increase, down to $5 \times 10^6$ MPN/100 mL, during the COVID-19 pandemic compared to the pre-COVID-19 pandemic period, but they remained at high levels during the post-COVID-19 pandemic phase, up to $6 \times 10^6$ MPN/100 mL (Figure 2). It seemed that bacterial loads' removal efficiency, during WWTP biological processes, was more significant, generating bigger gaps between influent and effluent bacterial loads during the pre-COVID-19 pandemic period compared to the COVID-19 pandemic and post-COVID-19 pandemic periods of time. Perhaps bacterial strains during the COVID-19 pandemic and post-COVID-19 pandemic acquired an antibiotic resistance which made them more resilient to WWTP removal treatments, and subsequently a higher bacterial amount ended up in effluent, which subsequently was released into the environment and natural streams.

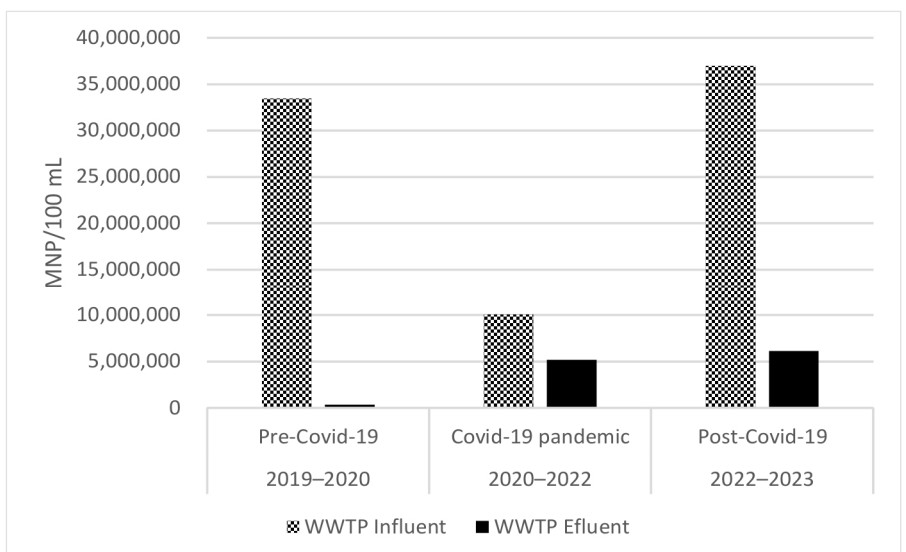

**Figure 2.** Total coliform bacteria load in WWTP influent and effluent between pre- and post-COVID-19 pandemic from a domestic WWTP in Bucharest, Romania. All results represented average values from at least two independent experiments. Each experiment was repeated twice ($p < 0.05$).

The presence of fecal populations, quantified in WWTP effluent, was around 50% of the total coliform's populations (Figure 3) during the pre- and post-COVID-19 pandemic with a peak of 55% detected during the COVID-19 pandemic. The percentage of fecal populations in the total coliforms population from influents spiked from around 20% during the pre-

and post-COVID-19 pandemics to up to 50% during the COVID-19 pandemic (Figure 3). Representatives of fecal groups such as *E. coli*, *Klebsiella* spp. were nominated, in recent studies, to have the highest frequency in acquiring antibiotic resistance genes [23,26] due to an enhanced antibiotic usage. This information matched our results linked to fecal population raised in total coliform bacterial strains during the COVID-19 pandemics compared to the pre- and post-COVID-19 pandemic periods (Figure 3). The percent of fecal bacterial detected in the total coliform bacterial population (Figure 3) isolated in WWTP influent matched the enterococcus bacterial population from the pre- to post-COVID-19 pandemic (Figure 4).

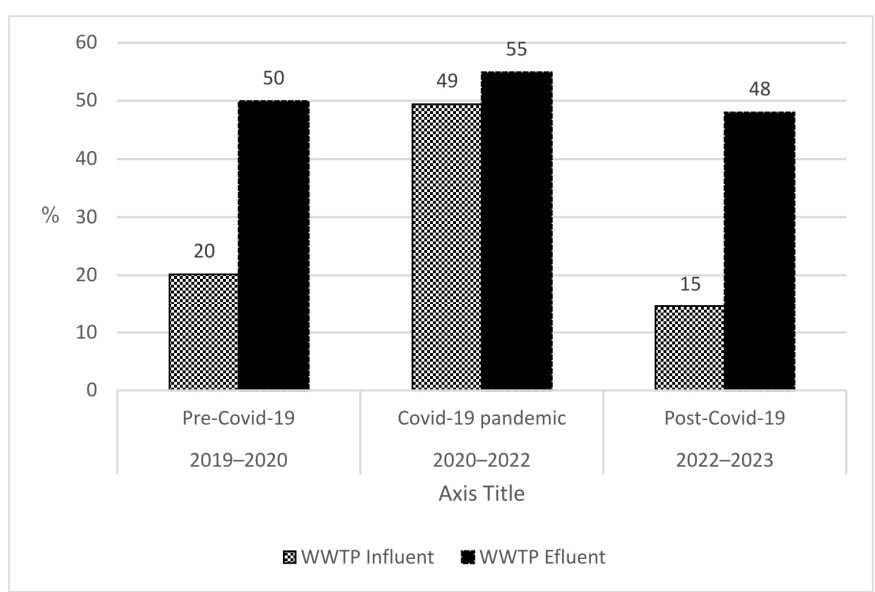

**Figure 3.** Percentage of fecal coliform bacteria from total coliform group in influent and effluent collected between the pre- and post-COVID-19 pandemic from a domestic WWTP in Bucharest, Romania. All results represent the average values from at least two independent experiments. Each experiment was repeated twice ($p < 0.05$).

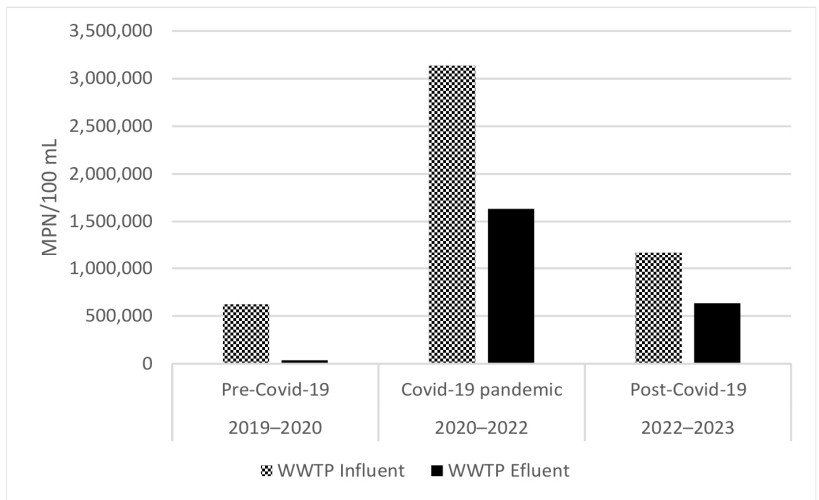

**Figure 4.** The efficiency of Bucharest WWTP on enterococci in the period covering pre- to post-COVID-19 pandemic—Bucharest, Romania. All results represent the average values from at least two independent experiments. Each experiment was repeated twice ($p < 0.05$).

The differences between influent and effluent enteroccoccus bacterial loads suggested that during the wastewater treatment steps, the bacterial loads were decreased up to 50% during the COVID-19 pandemic and post-COVID-19 pandemic periods of time, while in the

pre-COVID-19 pandemic, the enterococcus removal efficiency was up to 85% (Figure 4). The total coliform bacterial removal pattern during the wastewater treatment steps was similar to the enteroccocus from pre- to post-COVID-19 pandemics, although the bacterial removal WWTP efficiency was higher in the total coliform situation. The COVID-19 pandemic seemed to trigger the best premises for the development of highly resistant bacterial strains due to an increase in pharmaceutical compounds usage—especially biocides and antibiotics. Unfortunately, the antibacterial antibiotic resistance effect due to an increase in pharmaceutical compounds usage was more enhanced by an unregulated antimicrobial use and a lack of proper worldwide-accepted protocols. In addition, the mix between antibiotic and biocides generated an increase in antibiotic-resistant bacteria [27].

Enterococcus genus represents one of the most common bacterial presences in human infections; therefore, they are sensitive especially to gut microbiome changes induced by viral infections. Results showed that enterococcal infections increased during the COVID-19 pandemic [28] due to an increase in intestinal permeability induced by the viral infection.

### 3.3. Quantification of Surface Water Microbial Load

The COVID-19 pandemic lockdowns induced a significant decrease in coliforms density in Dambovita river upstream from the WWTP effluent evacuation site. The decrease in bacterial coliform in the aquatic environment was correlated to a significant reduced anthropic activity and with a significant increase in the use of pharmaceutical compounds, especial biocides and antibiotics. A recent Romanian study mentioned that the microbiological contamination decreased in uncontaminated surface waters from the southeastern part of Romania during the COVID-19 pandemic lockdown [29].

On the other hand, the spread of antibiotic resistance genes among bacterial strains together with the incomplete removal of antibiotics and biocides from wastewaters have had generated higher bacterial densities with a high potential to become antibiotic-resistant coliform bacteria. Larger amounts of these bacterial strains from WWTP effluent were discharged into the natural stream during the COVID-19 pandemic compared to the pre- and post-COVID-19 pandemic periods. Subsequently, there was a significant contamination of natural streams (upstream location from the WWTP effluent discharge) during the COVID-19 pandemic (Figure 5a).

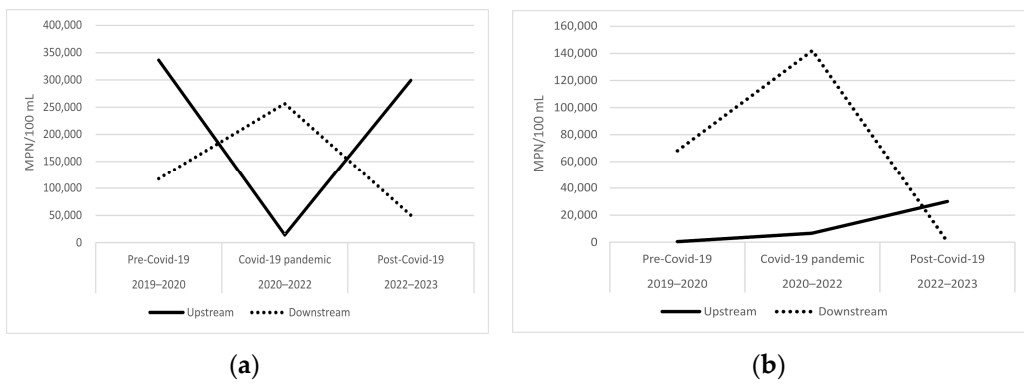

(**a**)                    (**b**)

**Figure 5.** Variation in the density of coliform bacteria (**a**) and enterococci (**b**) upstream and downstream of the Dambovita River covering the pre- to post-COVID-19 pandemic—Bucharest, Romania. All results represent the average values from at least two independent experiments. Each experiment was repeated twice ($p < 0.05$).

In addition to spreading antibiotic resistance genes to bacterial communities encountered in the Dambovita River's uncontaminated surface water, the bacterial strains from WWTP effluent found optimal conditions for multiplication, increasing the environmental bacterial contamination. Natural coliform bacterial densities from upstream the WWTP effluent discharge area (up to $3.5 \times 10^5$ MPN/100 mL) were very sensitive to pharmaceutical compounds because they were not in direct contact with massive concentrations of

antibiotics and biocides or antibiotic-resistant bacteria discharged in effluents by WWTPs. The downstream coliform bacterial density was about $1.1 \times 10^5$ MPN/100 mL due to the dilution factor of the effluent-receiving natural stream.

This pattern was observed for coliform bacteria during the pre- and post-COVID-19 pandemics when the coliform densities decreased up to 3.5-fold between upstream and downstream (Figure 5a). The uncontrolled use of biocides by population, during the COVID-19 pandemics, increased their presence in the environment and, therefore, they decreased the bacterial load from the natural upstream river by up to $0.2 \times 10^5$ MPN/100 mL. The low bacterial densities from the upstream river were very quickly increased by WWTP effluent bacterial discharge into the environment, reaching a density of $2.5 \times 10^5$ MPN/100 mL in the downstream river (Figure 5). WWTPs are hotspots for antibiotic resistance spreading by horizontal gene transfer; therefore, the bacterial load from WWTP effluent released into the environment not only overcame the environmental bacterial load (upstream) but also could convert the environmental bacteria into antibiotic-resistant bacteria. The enterococcal load released into the environment, through the WWTP effluent discharge, overcame the natural enterococcal populations. A quasi-absent upstream enterococcal load increased to $0.7 \times 10^5$ MPN/100 mL in the downstream river during the pre-COVID-19 pandemic (Figure 5b). The same growing of the enterococcal load from upstream ($0.1 \times 10^5$ MPN/100 mL) to downstream ($1.4 \times 10^5$ MPN/100 mL) in the river was also observed during the post-COVID-19 pandemic period of time. The same pattern was observed during the post-COVID-19 pandemic compared to the COVID-19 pandemic. This fact could be correlated with the fact that enterococci uncontrollably discharged into the environment became more resistant to antibiotics and biocides, and therefore increased their load upstream ($0.3 \times 10^5$ MPN/100 mL). The WWTP bacterial effluent load released into the environment made no difference in the overall downstream bacterial load, but the chemical load from WWTP effluents seemed to decrease the bacterial load that arrived from the upstream river (Figure 5b).

## 4. Conclusions

The Dambovita River flowing through Bucharest is an integral part of the city's urban life, and many industries as well as domestic activities are dependent on its water system. At the same time, various anthropic activities are a constant source of river contamination with coliform bacteria and enterococci.

The coronavirus generated a global pandemic with major sanitary and social changes, including more drastic lockdown periods throughout the world. The sanitary lockdown had a double effect on the environment. On one hand, it was beneficial for the environment [14] by improving the water quality of rivers and water bodies due to reduced anthropic activities such as industrial activities as well as reduced goods traffic and tourism transportation. On the other hand, lockdown put extra pressure on the environment, especially on the environmental microbiome by increasing the production and use of pharmaceutical antimicrobial compounds.

The COVID-19 pandemic had a direct impact on wastewater microbial loads from hospitals when the microbial population structure and dynamics were altered by a drastic decrease in coliform bacteria and enterococci populations. The same coliform reduction was observed for the WWTP influent during the COVID-19 pandemic. In spite of reducing the total coliforms load, the presence of fecal bacteria increased, which backed up the fact that the COVID-19 pandemic changed the bacterial population structure. Unfortunately, the decrease in the bacterial load from wastewaters and subsequently from natural streams was linked to a rise and dissemination of antibiotic resistance characteristics during the COVID-19 pandemic.

WWTPs have been hotspots for increasing the populations of antibiotic-resistant bacteria, especially during the COVID-19 pandemic, triggering a wide range of consequences such as contamination of wastewater treatment plant workers [30]. In addition, their contamination with antibiotic-resistant bacteria could increase the chances of nosocomial

diseases generation once these workers or their first contact relatives come in contact with healthcare units. Moreover, the activated sludge from WWTP has been used as fertilizer for agricultural activities; thus, the antibiotic-resistant bacteria could also spread into the environment [31]. Not only would antibiotic-resistant bacteria spread into the environment via WWTP effluents released into the natural stream: along with them, other chemical compounds could spread into environmental biota with dire consequences over a long period of time [32].

Monitoring the microbiological parameters from domestic and industrial wastewater, WWTP biological treatment steps and the environment has been very useful in predicting the potential threats against human and environmental health.

Overall, the COVID-19 pandemic had a long-term impact on microbiological parameters from hospitals to the environment, since most of the microbiological changes were still detected one year after the post-COVID-19 pandemic.

**Author Contributions:** Conceptualization, A.R.B. and M.N.-L.; Formal analysis A.R.B., D.M.R., C.S., S.G. and I.L.; Writing—original draft, A.R.B., L.F.P., F.V.C., L.N., C.M. and M.N.-L.; Writing—Review and Editing, A.R.B. and L.F.P. All authors have read and agreed to the published version of the manuscript.

**Funding:** This study was supported by the PN-III-P4-ID-PCCF-2016-0114 research project awarded by UEFISCDI.

**Institutional Review Board Statement:** Not applicable.

**Informed Consent Statement:** Not applicable.

**Data Availability Statement:** The data presented in this study are available on request from the corresponding author. The data are not publicly available due to ongoing research in this field.

**Acknowledgments:** This work was "Nucleu" Program within the National Research Development and Innovation Plan 2019–2022 as well as 2022–2027 with the support of Romanian Ministry of Research, Innovation and Digitalization, contract no. 20N/2019, Project code PN 19 04 02 01 and contract no. 3N/2022.

**Conflicts of Interest:** The authors declare no conflicts of interest.

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
