# Peer review of "The COVID-19 Pandemic Impact of Hospital Wastewater on Aquatic Systems in Bucharest"

_water, doi:10.3390/w16020245_

Round 1
Reviewer 1 Report
Comments and Suggestions for Authors
Title: The title is precise but somewhat uninformative. I suggest something like “Microbiological Impact of Hospital Wastewater on Streams in Bucharest during COVID-19.”
General: I don’t understand the use of the term “emissaries.” I believe the authors mean “streams” or “rivers” and this is just a matter of translation.
General: English needs work. I generally understand but the syntax, punctuation, SVA, and other elements need editing.
Results: The Results section should contain text, not just figures. Alternatively Results and Discussion in one section may be appropriate.
Results: Color figures would be appreciated.
General: The authors capture the critical 1–3 year post-Covid period suggested by Sowby and Lunstad (Considerations for Studying the Impacts of COVID-19 and Other Complex Hazards on Drinking Water Systems,” https://doi.org/10.1061/(ASCE)IS.1943-555X.0000658). This reference could justify the authors’ work. Another one is “Water and wastewater systems and utilities: Challenges and opportunities during the COVID-19 pandemic” (https://doi.org/10.1061/(ASCE)WR.1943-5452.0001373) by Berglund et al. Both articles further comment on the disruptions the authors describe in the Discussion.
Conclusions: I appreciate the brevity of the paper. However, I feel that an obvious point is missing: hospital wastewater during COVID adversely affected the microbiological communities of Bucharest’s rivers. The authors should emphasize this conclusion and discuss the implications.
Comments on the Quality of English Languagesyntax, punctuation, SVA, and other elements
Author Response
Dear Reviewer 1
Thank you for your comments which helped us to improve the manuscript.
We made the following modifications / corrections:
1.1. Title was changed
1.2. The term emissaries was clarified even from the abstract : “…Rivers flowing through cities are natural emissaries for WWTP discharges...”
1.3. Englished was improved
1.4. The results and discussion sections were merged
1.5. We prefer black and white figures since colors could look a bit different when printed
1.6. More information was added regarding microbiological parameters and their importance/implication on human and environmental health from Romania, please see the discussion section (overall, references were extended from 24 to 32)
Please, read the attach paper.

Reviewer 2 Report
Comments and Suggestions for Authors
Pharmaceuticals are biologically active compounds; therefore, their presence in the environment, evening race amounts, can negatively affect the state of the aquatic ecosystem. In this regard, information on the pharmaceuticals’ release from WWTPs could help to predict the list of target compounds in the environment. Thus, the problem considered in the article is very relevant. However, the manuscript suffers from a number of shortcomings.
1. Instead of presenting the results, the authors limited themselves to presenting several figures. No statistical analysis available. Are the values shown in the graphs averages? If there were two independent experiments with each experiment repeated twice, where is the data on the dispersion or the max/min spread?
2. The graphs are done carelessly. The schematic map is not readable.
3. It is advisable to discuss the possibilities of increasing the efficiency of these treatment facilities to reduce bacterial pollution of the environment.
I hope that the authors will easily cope with eliminating the shortcomings, which will help improve the article.
Author Response
Dear Reviewer 2,
Thank you for your comments helped us to improve the manuscript.
We made the following modifications / corrections:
2.1. Clarification on results data and statistical analysis was added especially on Material and Methods section as well as on figure legends.
2.2. We tried to improve the quality of the figures, please let us know if we need to do further work on them. We feel that colored figure could bring more issues than answers.
2.3. Discussion was improved with more information of the microbiological impact on humans (WWTP workers), agriculture and environment.
Please, read the attached paper.

Reviewer 3 Report
Comments and Suggestions for Authors
1. Abstract, many background in present. More detail of the results and conclusion are needed, especially the “impact” and “structural changes”.
2. All results are from at least two independent experiments with each experiment repeated twice. The method of statistical analysis would be listed in chapter 2.
3. Results are only consisted of several figures, general description of the results are needed. It may combine the result and discussion with some sub-titles.
Comments on the Quality of English LanguageThe English writing is readable.
Author Response
Dear Reviewer 3,
Thank you for your comments helped us to improve the manuscript.
We made the following modifications / corrections:
3.1. Abstract and discussion sections were improved
3.2. Clarification on results data and statistical analysis was added especially on Material and Methods section as well as on figure legends.
3.3. The results and discussion sections were merged
Please read the attached paper.

Round 2
Reviewer 1 Report
Comments and Suggestions for Authors
The use of “emissary” throughout the paper is still unusual. Even though defines, this is not the right English term.
English is still generally unsuitable. For example, a better title syntax is “The COVID-19 pandemic impact of hospital wastewater on aquatic systems in Bucharest”
Comments on the Quality of English LanguageSee above.
Author Response
Dear reviewer,
Thank you for your comments.
We changed all the “emissary” words to “stream”. We agreed that your proposed titled could bring more clarity impact. Other English minor typos and grammar mistakes were corrected.
All changes are highlighted in yellow.

Reviewer 2 Report
Comments and Suggestions for Authors
The authors significantly improved the manuscript and corrected the indicated comments and recommendations. It is likely that color drawings will better illustrate the results and sampling map.
The article may be published in the Journal Water and will be of interest to a wide range of readers.
Author Response
Dear reviewer,
Thank you for your opinions which we also think we’ll cover a broad range of readers.
We made some changes highlighted in yellow.

Reviewer 3 Report
Comments and Suggestions for Authors
1. 2.3. Statistically analysis. The statistical data was obtained by ANOVA tests. Normal 95% confidence value (P<0.05) is requested. P<0.05 is enough not 0.02 in the legends of figures,
2. Subtitles of the results and discussion are needed.
Author Response
Dear reviewer
Thank you for your comments.
Indeed , p< 0.05 is enough and reflects the overall reality of these results, so we kept p<0.05 all over the text.
It was a great idea to add subtitles to facilitate the understanding of the paper. Three more subtitles were added to results and discussion section: 3.1. Quantification of hospital wastewater microbial load, 3.2. Microbial load from WWTP, 3.3. Quantification of surface water microbial load.
All changes are highlighted in yallow.
Other English minor typos and grammar mistakes were corrected.
